# Local data for local programming: Results from an HIV biobehavioral survey among people who inject drugs in Livingstone, Lusaka, and Ndola, Zambia, 2021

Daniel Woytowich[1☯*], Anne F. McIntyre[1☯], Hiwote Solomon[1☯¤], Brave Hanunka[2‡], Lazarus Chelu[3‡], Tepa Nkumbula[3‡], Leigh Tally[2‡], Ray Handema[4‡], Shepherd Khondowe[4‡], Kelvin Kapungu[4‡], Lophina Chilukutu[3‡], Innocent Bwalya[4‡] Chipili Mulemfwe[3‡], Melvin Mwansa[3‡], Kennedy Mutale[5‡], Neena M. Philip[6‡], Giles Reid[6‡] Joyce J. Neal[7‡], Maria Lahuerta[6‡], Lauren E. Parmley[6‡], Hannah Chung[6‡], Avi J. Hakim[1‡¤], Jonas Z. Hines[2‡] Evelyn Kim[1¤], John Mwale[8‡], Lloyd B. Mulenga[8,9‡]

1 Division of Global HIV and TB, U.S. Centers for Disease Control and Prevention, Global Health Center, Atlanta, Georgia, United States of America, 2 Division of Global HIV and TB, U.S. Centers for Disease Control and Prevention, Global Health Center, Lusaka, Zambia, 3 ICAP at Columbia University, Mailman School of Public Health, Lusaka, Zambia, 4 Tropical Disease Research Centre, Ndola, Zambia, 5 Key Populations Consortium, Lusaka, Zambia, 6 ICAP at Columbia University, Mailman School of Public Health, New York, New York, United States of America, 7 Division of Global HIV and TB, U.S. Centers for Disease Control and Prevention, Global Health Center, Phnom Penh, Cambodia, 8 Ministry of Health Zambia, National HIV/AIDS/STI/TB Council, Lusaka, Zambia, 9 Division of Infectious Diseases, University of Zambia, School of Medicine, Lusaka, Zambia

☯ These authors contributed equally to this work.
‡ These authors also contributed equally to this work.
¤ Current Address: U.S. Centers for Disease Control and Prevention, Global Health Center, Division of Global HIV and TB, Atlanta, Georgia, United States of America
* toj2@cdc.gov

## Abstract

### Introduction

People who inject drugs (PWID) in Zambia are an understudied population at high risk for HIV acquisition and transmission. We report here on the progress within the PWID communities of Livingstone, Lusaka, and Ndola, Zambia towards the Joint United Nations Programme on HIV/AIDS (UNAIDS) 95-95-95 targets.

### Methods

A biobehavioral survey used respondent-driven sampling to survey 235 PWID in Livingstone, 349 in Lusaka, and 259 in Ndola in 2021–22. Questions on HIV and injection drug use were administered, and blood was collected for HIV, syphilis, Hepatitis B, and Hepatitis C testing. Weighted prevalence and 95% confidence intervals (CIs) were calculated using Gile's sequential sampling estimator.

**Data availability statement:** The underlying data for this study are owned by the Zambian Ministry of Health and the National AIDS Council. Due to the sensitive nature of the data, which involve key populations that face legal and social risks, there are strict ethical and legal restrictions on data sharing. These restrictions are mandated by the institutional review board (IRB) requirements to protect participant privacy and confidentiality. For these reasons, the de-identified data can only be accessed upon request and after approval by the relevant Zambian health authorities. Researchers interested in accessing the data should contact Dr. Kebby Chongwe Musokotwane, the Director General of the National HIV/AIDS/STI/TB Council [KMusokotwane@nacsec.org.zm; +260-211-255044]. Requests will be evaluated to ensure compliance with ethical standards and participant safety."

**Funding:** This activity was funded by the US President's Emergency Plan for AIDS Relief (PEPFAR) (https://www.state.gov/pepfar/) through the US Centers for Disease Control and Prevention (CDC) under the terms of cooperative agreement "Supporting sustainable surveillance systems among key populations and support the Government of Zambia to improve HIV-related services for KPs" (Prime Award No. 1NU2GGH002056). The findings and conclusions in this report are those of the author(s) and do not necessarily represent the official position of the funding agencies. CDC employees, whose specific roles are outlined in the author contributions section, provided technical assistance and input into the study design, data collection, data analysis, decision to publish, and preparation of the manuscript."\

**Competing interests:** The authors have declared that no competing interests exist.

## Results

In Livingstone, Lusaka, and Ndola, HIV prevalence among PWID was 11.9% (95% CI: 7.3, 16.5), 7.3% (95% CI: 4.5, 10.2), and 21.9% (95% CI: 14.5, 29.3), respectively. Among HIV-positive PWID in Livingstone, 70.7% (95% CI: 55.4, 85.0) were aware of their HIV status (95% is 1st UNAIDS target), 100% of those were on antiretroviral therapy (ART) (95% is 2nd UNAIDS target), and 100% of those achieved viral load suppression (VLS) (95% is 3rd UNAIDS target). In Lusaka, 66.0% (95% CI: 49.3, 82.2) were aware, 75.7% (95% CI: 51.1, 99.9) were on ART, and 66.3% (95% CI: 42.1, 90.9) achieved VLS. In Ndola, 60.2% (95% CI: 44.1, 76.0), 100%, and 90.2% (95% CI: 82.2, 98.3) were aware, on ART, and achieved VLS, respectively.

## Conclusions

Awareness of HIV status was low among PWID living in Livingstone, Lusaka, and Ndola, Zambia. Treatment and VLS progress were lacking in Lusaka and Ndola as well with Lusaka showing the least progress toward all three UNAIDS targets. Our site-level findings highlight critical gaps in PWID-specific HIV awareness, treatment, and VLS status in three major urban areas in Zambia that limit progress toward HIV epidemic control in this hard-to-reach population.

## Introduction

People who inject drugs (PWID) are especially susceptible to acquiring HIV and accounted for approximately 8.0% of new HIV infections globally in 2022 [1–6]. A major reason for this is the sharing of used needles [1,7–9]. Other factors include social isolation, poor access to health care, stigma, discrimination, and criminalization [2,7,10–14]. Whereas the world's general population of 15–49-year-olds have an HIV prevalence of 0.6% (95% CI: 0.6, 0.7) [15], 15–64-year-old PWID worldwide have an HIV prevalence of 15.2% (95% CI: 10.3, 20.9) [11]. In Sub-Saharan Africa (SSA) the HIV prevalence among the general population of 15–49-year-olds is 3.4% (95% CI: 3.1, 3.8) [15], compared to 11.2% (95% CI: 5.4, 19.0) among 15–64-year-old PWID [11]. HIV transmission occurs between PWID when injecting equipment is shared and from PWID to others through activities like sex work [16], in which 19.6% (95% CI: 11.1, 31.3) of SSA PWID have recently (within a year) participated [11].

The Joint United Nations Programme on HIV/AIDS (UNAIDS) 95-95-95 targets [17] are intended to be achieved by 2030 [17,18]. The first target is for 95% of people living with HIV (PLHIV) to be aware of their HIV status. The second is for 95% of PLHIV who are aware of their serostatus to receive sustained antiretroviral therapy (ART), and the third is for 95% of those on ART to achieve viral load (VL) suppression (VLS) [17,18], defined as HIV RNA of <1000 copies/mL during the most recent VL test [17–19]. UNAIDS 95-95-95 goals involve reaching these benchmarks not only in the general population, but in priority subpopulations [17] like PWID as well.

In 2021, estimates for the 1st, 2nd, and 3rd 95-95-95 target proportions in Zambia's general population were 88.7%, 98.0%, and 96.3%, respectively, indicating that the 2nd and 3rd goals have already been met [20]. However, progress toward the 1st target lags behind by over six percentage points [20]. Although there is some limited information about injection drug use and HIV in SSA [21–25], specific data on the status of the HIV epidemic within Zambia's PWID community are lacking [13]. Of note, the injection of narcotic and psychotropic substances, along with their precursor chemicals is illegal in Zambia. In addition, there are no PEPFAR or government-sponsored Opioid Agonist Maintenance Therapy (OAMT) or Needle and Syringe Programs (NSPs) available for the treatment of stimulant dependence in Zambia. There is no published information on HIV awareness, ART uptake, and/or VLS among Zambian PWID communities. Elucidating current progress toward 95-95-95 targets among PWID in Zambia is essential for understanding the performance of current outreach and interventions to this group [26–28], and how those might be improved for greater impact.

In 2021, Zambia implemented the first ever probability-based biobehavioral survey (BBS) to estimate HIV prevalence, HIV-related indicators, syphilis, Hepatitis B, and Hepatitis C prevalence among PWID. Here, we report site-specific progress toward reaching the UNAIDS 95-95-95 targets among PWID in Livingstone, Lusaka, and Ndola, Zambia. Socio-demographic variables, injecting practices, sexual behaviors, stigma, discrimination, risk perception, and service use that may have influenced the 95-95-95 results were examined as well. Our goal was to analyze and interpret information at the local level, rather than at the national or provincial level, to ensure that any nuanced differences across survey sites would not be masked in our results and therefore possibly overlooked in future program design.

## Methods

### Study design and eligibility criteria

Three survey sites (Livingstone, Lusaka, and Ndola) were selected based on a previous formative assessment of persons who use drugs in Zambia [13]. In the six months prior to the survey, the consensus estimate among 15–64-year-olds for the PWID population size in Livingstone, Lusaka, and Ndola was 0.9% (95% CI: 0.7, 1.5), 0.2% (95% CI: 0.1, 0.5), and 0.6% (95% CI: 0.4, 0.7), respectively. Livingstone is in Southwest Zambia on the border with Zimbabwe and has a high influx of visitors for Victoria Falls, but nonetheless has the smallest population of the three survey sites. Lusaka is in Central Zambia, is the nation's capital, and has the largest population of the three sites. Ndola lies roughly 300 km to the north of Lusaka near the border with the Democratic Republic of the Congo. PWID in Livingstone, Lusaka, and Ndola were recruited using respondent-driven sampling (RDS), a chain referral method used for accessing hard-to-sample populations and approximating a probability sample in the absence of a sampling frame [29,30]. In 2016–17, the proportion of 15–59-year-old PLHIV with VLS in Zambia was 59.2% [20]. Assuming a design effect of 2 and nonresponse rate of 5%, a sample size of 195 HIV-positive participants was needed for a two-sided 95% confidence interval (CI) ranging from 50.0% to 70.0% when the proportion of HIV-positive people with VLS was 60.0%. Therefore, a target sample size of 780 PWID for each city was calculated to obtain a sample size of 195 HIV-positive PWID participants per city assuming an HIV prevalence of 25%. However, this was determined to be unattainable after consultation with local stakeholders supporting PWID in Zambia due to workforce and funding limitations. Therefore, this target was divided across the sites based upon the size of the PWID population in each city (Livingstone, n = 215; Lusaka, n = 350; Ndola, n = 215).

PWID were eligible if they were 16 years or older; provided verbal informed consent; spoke English, Chinyanja, Cibemba, Silozi, Kikaonde, or Chitonga; lived in the survey city; self-reported injecting drugs for non-medical purposes in the past three months; had a valid survey coupon; and did not previously participate in the survey. Six to eight PWID seeds began recruitment of peers. Seeds were purposively selected based on eligibility for the survey, large social networks, enthusiasm and support for the survey, and a diverse range of sociodemographic characteristics. Successive recruitment waves were carried out until the desired sample size was reached. Recruitment occurred from November 2021 to February 2022.

## Survey process and data collection

During the first of two survey visits, candidate participants presented recruitment coupons and were screened for eligibility. Verbal informed consent was obtained for the questionnaire, biomarker testing, and storage of blood samples. Staff administered tablet-based in-person interviews using standardized instruments. Interview domains included: demographics, sexual history, injection drug use history, HIV knowledge, uptake of HIV and sexually transmitted infection (STI) services, history of STIs, social cohesion, stigma, and COVID-19. The two-item Patient Health Questionnaire (PHQ-2) was used to screen for depression and the Alcohol Use Disorders Identification Test (AUDIT) for alcohol disorders [31,32]. Participants were tested for HIV, HIV VL, active syphilis, hepatitis B virus (HBV), and hepatitis C virus (HCV) as described below, received pre-and post-test counseling along with their rapid test results, and scheduled a second visit. Referrals to care were provided to those who tested positive for HIV, active syphilis, HBV, and HCV. Participants who had STI symptoms were also referred to care. Approximately two weeks later, participants returned for the second visit to provide information about the number and characteristics of peers they recruited. Persons who tested positive for HIV received HIV VL test results. Participants received $13.00 and $5.00 reimbursement for their first and second visits, respectively; along with $3.00 for each peer they referred that completed a survey. Coupon distribution was monitored using an RDS coupon management spreadsheet and ceased when 95% of the sample size was reached, whereas overall recruitment monitoring occurred weekly until both sample size and convergence were reached.

## Laboratory and testing procedures

Participants who consented to being tested received pre-test counseling for HIV, active syphilis, HBV, and HCV that followed national guidelines [33]. Venipuncture was used to collect 15 mL of venous blood into two 7.5 mL EDTA tubes for HIV, HIV VL, active syphilis, HBV, and HCV testing. Rapid testing for HIV, syphilis, HBV, and HCV was performed at the survey site using venous whole blood from the first 7.5 mL tube. Participants with a reactive result on both screening (Determine™ HIV-1/2 [Abbott Molecular Inc., Des Plaines, Illinois, United States]) [34] and confirmatory HIV tests (SD Bioline™ HIV-1/2 [Abbott Molecular Inc., Des Plaines, Illinois, United States]) [35] were classified as HIV-positive. Samples reactive on the initial test but nonreactive on the confirmatory test were retested, and results were returned immediately. The remaining blood and second 7.5 mL tube were then centrifuged at 3000 revolutions per minute to separate into plasma which was stored at -20°C until it was shipped to the Tropical Diseases Research Centre (TDRC) laboratory in Ndola by a trained driver at room temperature, which occurred weekly. Plasma was stored at −20°C in the TDRC laboratory until it was tested, within one week of arrival, then kept at −80°C for long-term storage. Laboratory-based testing was conducted at TDRC for active syphilis, HIV VL, and HIV recency classification per the recent infection testing algorithm. Confirmation of positive rapid assays were conducted for HIV, HCV, and syphilis. Participants with discordant results in the confirmatory test were tested again during their second visit. For those classified as HIV-positive, HIV-1 VL testing was performed using the COBAS AmpliPrep/COBAS TaqMan™ HIV-1 Test, v2.0 (Roche Molecular Diagnostics, Indianapolis, Indiana, United States) [36]. Active syphilis testing was conducted using the Chembio™ DPP Syphilis Screen and Confirm Assay (Medford, New York, United States) [37], with confirmatory testing being done with the SD Bioline™ Syphilis 3.0 kit (Abbott Molecular Inc., Chicago, Illinois, United States) [38]. Acute and chronic HBV was tested for with the Determine™ HBsAg test (Abbott Molecular Inc., Chicago, Illinois, United States) [39]. Testing for current or past HCV infection was conducted with the SD Bioline™ HCV test (Abbott Molecular Inc., Chicago, Illinois, United States) [40] whereas confirmatory testing was conducted with the Applied Biosystems™ 7500 RT PCR platform (Waltham, Massachusetts, United States) [41] and the Applied Biosystems Taqman™ Fast Virus 1-Step Master Mix for qPCR (Waltham, Massachusetts, United States) [42].

## Data management and analysis

RDS Analyst software (Los Angeles, California, United States) [43], SAS software (Cary, North Carolina, United States) [44], and R software [45], was used for data management, weighting, and analysis. Weighted prevalence and 95% CIs

for all variables were calculated using Gile's sequential sampling estimator [46]. Participants were categorized as being aware of their HIV status (1st 95 target) if they reported a prior HIV diagnosis and/or had a VL of <200 copies/mL. Of those, PWID were classified as on ART (2nd 95) based on self-report of current ART use and adjusted for a VL of <200 copies/mL [47]. VLS (3rd 95) was defined as having a VL of <1000 copies/mL.

## Human subjects considerations

This survey was conducted by the Zambian National HIV/AIDS/STI/TB Council (NAC), the Zambian Ministry of Health (MoH), ICAP at Columbia University, and the Tropical Diseases Research Centre (TDRC). It was funded by the United States (US) President's Emergency Plan for AIDS Relief (PEPFAR) and supported by the US Centers for Disease Control and Prevention (CDC). The survey protocol was reviewed and approved by the Columbia University Medical Center Institutional Review Board, TDRC ethics committee, Zambia MoH, and National Health Research Authority. This activity was also reviewed by CDC, deemed not research, and was conducted consistent with applicable US federal law and CDC policy (45 C.F.R. 46). The NAC was interested in including 16–17-year-olds in this study due to high incidence of HIV in this population and lack of data on them. Since the legal age of consent is 16 years in Zambia, people of this age group were included after providing consent. Verbal informed consent on procedures, potential risks, benefits, and how to report concerns was obtained from all participants and was witnessed by staff trained in human subject protections. Verbal, as opposed to written consent, was obtained since this survey met both conditions of 45CFR46.117(c) in that (1) the signed consent would be the only record that could link respondents to the survey which would increase the risk of breach of confidentiality, and (2) the survey presents no more than minimal risk of harm to respondents and involves no procedures for which written consent is normally required outside of the research context. After verbal consent was given, it was documented by the interviewer signing on behalf of the participant. All participation was confidential. CDC investigators had no interaction with participants and no access to identifiable data or specimens.

## Results

### Recruitment, screening, enrollment, and testing

Six seeds were used in Livingstone and Ndola, whereas eight were used in Lusaka (Table 1). Mean number of recruits per seed ranged from 38.2–57.0. The longest wave was 18 (Lusaka). Coupon return rate ranged from 48.2%-53.8%; and 249, 479, and 303 PWID were screened in Livingstone, Lusaka, and Ndola, respectively. Of those, 235, 349, and 259 were eligible, enrolled, and tested for biomarkers. Of those, 176, 253, and 184 participants returned for a second visit.

### Sociodemographic and sexual history characteristics

**Livingstone.** The median age of PWID in Livingstone was 23.0 years (interquartile range [IQR]: 20.0–30.0) (Table 2). The majority were male (83.6%, 95% CI: 73.4, 93.6), never married (76.3%, 95% CI: 70.3, 82.3), had no children (59.2%,

**Table 1. Recruitment, screening, enrollment, and testing characteristics—Livingstone, Lusaka, and Ndola, Zambia, 2021.**

|  | Livingstone | Lusaka | Ndola |
|---|---|---|---|
| **Number of seeds** | 6 | 8 | 6 |
| **Mean number of recruits by seed** | 38.2 | 57.0 | 42.2 |
| **Length of longest wave** | 6 | 18 | 8 |
| **Percent of coupons returned** | 48.2% | 51.5% | 53.8% |
| **Number of PWID screened** | 249 | 479 | 303 |
| **Number (%) of PWID eligible out of those screened** | 235 (94.4) | 349 (72.9) | 259 (85.5) |
| **Number (%) of PWID enrolled out of those eligible** | 235 (100.0) | 349 (100.0) | 259 (100.0) |
| **Number (%) tested for biomarkers out of those enrolled** | 235 (100.0) | 349 (100.0) | 259 (100.0) |
| **Number (%) returned for second visit out of those enrolled** | 176 (74.9) | 253 (72.5) | 184 (71.0) |

**Table 2. Sociodemographic and sexual history characteristics of PWID—Livingstone, Lusaka, and Ndola, Zambia, 2021.**

| | Livingstone | | | Lusaka | | | Ndola | | |
|---|---|---|---|---|---|---|---|---|---|
| **Sample Size** | **n = 235** | | | **n = 349** | | | **n = 259** | | |
| | **Sample** | | **Population** | **Sample** | | **Population** | **Sample** | | **Population** |
| | **n** | **%** | **% (95% CI)** | **n** | **%** | **% (95% CI)** | **n** | **%** | **% (95% CI)** |
| **Median Age (IQR), years** | 23.0 (20.0, 30.0) | | | 24.0 (21.0, 28.0) | | | 28.0 (23.0, 32.0) | | |
| **Age Categories, years** | | | | | | | | | |
| 16-24 | 135 | 57.4 | 57.7 (48.6-66.9) | 172 | 49.3 | 51.2 (43.8-58.5) | 80 | 30.9 | 30.4 (21.5-39.1) |
| 25-34 | 72 | 30.6 | 30.9 (23.0-38.8) | 149 | 42.7 | 41.6 (34.6-48.6) | 124 | 47.9 | 50.8 (42.3-59.3) |
| 35 and over | 28 | 11.9 | 11.3 (7.0-15.7) | 28 | 8.0 | 7.3 (3.6-11.0) | 55 | 21.2 | 18.9 (11.6-26.2) |
| **Sex** | | | | | | | | | |
| Male | 199 | 84.7 | 83.6 (73.4-93.6) | 335 | 96.0 | 97.4 (96.0-98.9) | 173 | 66.8 | 65.9 (56.0-75.9) |
| Female | 36 | 15.3 | 16.4 (6.3-26.6) | 13 | 3.7 | 2.4 (1.0-3.8) | 77 | 29.7 | 29.8 (20.6-39.1) |
| **Marital Status** | | | | | | | | | |
| Never Married | 178 | 75.7 | 76.3 (70.3-82.3) | 240 | 68.8 | 70.3 (63.5-77.4) | 151 | 58.3 | 58.0 (49.0-66.8) |
| Married | 20 | 8.5 | 7.3 (4.1-10.6) | 33 | 9.5 | 7.8 (4.1-11.6) | 46 | 17.8 | 17.6 (10.9-24.3) |
| No longer Married | 37 | 15.7 | 16.4 (11.3-21.5) | 76 | 21.8 | 21.8 (16.0-28.0) | 62 | 23.9 | 24.4 (16.8-32.1) |
| **Number of living children** | | | | | | | | | |
| None | 132 | 56.2 | 59.2 (50.9-67.6) | 164 | 47.0 | 54.4 (46.7-62.1) | 96 | 37.1 | 37.6 (28.6-46.6) |
| 1 | 60 | 25.5 | 23.1 (17.1-29.0) | 128 | 36.7 | 31.0 (24.2-36.8) | 83 | 32.0 | 30.7 (22.4-38.9) |
| 2 or more | 43 | 18.3 | 17.7 (11.6-23.9) | 57 | 16.3 | 15.1 (9.9-20.3) | 80 | 30.9 | 31.8 (23.4-40.2) |
| **Highest level of education completed** | | | | | | | | | |
| None | 13 | 5.5 | 6.7 (1.3-12.1) | 25 | 7.2 | 7.3 (3.4-11.3) | 0 | 0 | – |
| Primary | 91 | 38.7 | 34.8 (27.5-42.2) | 103 | 29.5 | 32.7 (26.4-39.1) | 34 | 13.1 | 11.9 (6.3-17.5) |
| Secondary or higher | 131 | 55.7 | 58.4 (50.7-66.1) | 221 | 63.3 | 60.0 (53.1-66.8) | 224 | 86.5 | 87.7 (82.1-93.4) |
| **Homeless** | 8 | 3.4 | 2.5 (0.6-4.4) | 40 | 11.5 | 13.2 (7.8-18.6) | * | * | * |
| **Median age at first sex (IQR), years** | 15.0 (13.0, 17.0) | | | 16.0 (15.0, 18.0) | | | 17.0 (16.0, 19.0) | | |
| **Age at first sex, years** | | | | | | | | | |
| <15 | 102 | 43.4 | 42.0 (34.1-49.8) | 76 | 21.8 | 18.4 (13.6-23.2) | 53 | 20.5 | 22.2 (15.2-29.3) |
| 15-19 | 115 | 48.9 | 50.1 (42.4-58.0) | 235 | 67.3 | 67.7 (61.5-73.9) | 155 | 59.8 | 56.8 (48.9-64.7) |
| ≥ 20 | 18 | 7.7 | 7.9 (3.8-11.9) | 38 | 10.9 | 13.0 (7.4-20.2) | 51 | 19.7 | 21.0 (10.5-27.7) |
| **Lifetime number of sexual partners (vaginal and/or anal sex)** | | | | | | | | | |
| 1 | 9 | 3.8 | 2.9 (0.7-5.1) | 19 | 5.4 | 6.1 (3.1-9.0) | * | * | * |
| 2 | 19 | 8.1 | 8.0 (3.9-12.1) | 27 | 7.7 | 9.2 (4.6-14.0) | 13 | 5.0 | 8.1 (3.2-12.9) |
| 3-5 | 64 | 27.2 | 25.1 (18.4-31.7) | 120 | 34.4 | 35.6 (29.1-42.3) | 68 | 26.3 | 24.3 (17.4-31.2) |
| 6 or more | 133 | 56.6 | 60.2 (52.8-67.6) | 172 | 49.3 | 46.0 (39.1-52.6) | 172 | 66.4 | 66.5 (58.4-74.6) |
| **Ever forced to have sex** | 37 | 15.7 | 15.8 (9.2-22.4) | 32 | 9.2 | 9.1 (5.4-12.7) | 96 | 37.1 | 35.9 (28.2-43.7) |
| **Number of people whom you sold vaginal and/or anal sex to in the last 6 months[I]** | | | | | | | | | |
| None | 211 | 89.8 | 89.9 (84.6-95.2) | 338 | 96.8 | 97.6 (95.9-99.3) | 175 | 67.6 | 70.6 (62.5-78.7) |
| 1-2 | 7 | 3.0 | 3.3 (0.3-6.2) | 9 | 2.6 | 2.1 (0.4-3.8) | 31 | 12.0 | 10.0 (5.3-14.6) |
| 3 or more | 17 | 7.2 | 6.8 (2.1-11.5) | * | * | * | 43 | 16.6 | 14.9 (8.8-21.1) |

Population estimates and 95% Confidence Intervals are weighted using Gile's SS estimator, whereas sample estimates are unweighted. CI: Confidence Interval. IQR: Interquartile Range. *Suppressed due to number being ≤ 5. [I]If one sold sex prior to the age of 18, they were a sexually exploited minor.

95% CI: 50.9, 67.6), completed at least a secondary education (58.4%, 95% CI: 50.7, 66.1), and 2.5% (95% CI: 0.6, 4.4) were homeless. Median age at first sex was 15.0 years (IQR: 13.0–17.0), 60.2% (95% CI: 52.8, 67.6) had six or more lifetime vaginal and/or anal sexual partners, 15.8% (95% CI: 9.2, 22.4) had been forced to have sex in their lifetime, and

6.8% (95% CI: 2.1, 11.5) sold sex, or were a sexually exploited minor, to at least three people in the six months preceding the survey.

**Lusaka.** Median age of PWID in Lusaka was 24.0 years (IQR: 21.0–28.0) and 97.4% were male (95% CI: 96.0, 98.9). The majority were never married (70.3%, 95% CI: 63.5, 77.4), had no children (54.4%, 95% CI: 46.7, 62.1), 60.0% completed at least a secondary education (95% CI: 53.1, 66.8), and 13.2% (95% CI: 7.8, 18.6) were homeless. Median age at first sex was 16.0 years (IQR: 15.0–18.0), 46.0% (95% CI: 39.1, 52.6) had six or more lifetime sexual partners, 9.1% (95% CI: 5.4, 12.7) had been forced to have sex, and 2.1% (95% CI: 0.4, 3.8) sold sex, or were a sexually exploited minor, to 1–2 people in the last six months.

**Ndola.** Median age of PWID in Ndola was 28.0 years (IQR: 23.0–32.0) and almost a third (29.8%, 95% CI: 20.6, 39.1) were female. Most were never married (58.0%, 95% CI: 49.0, 66.8), 31.8% had two or more children (95% CI: 23.4, 40.2), and 87.7% completed secondary education or higher (95% CI: 82.1, 93.4). Median age at first sex was 17.0 years (IQR: 16.0–19.0), 66.5% (95% CI: 58.4, 74.6) had six or more lifetime sex partners, 35.9% had been forced to have sex (95% CI: 28.2, 43.7), and 14.9% (95% CI: 8.8, 21.1) sold sex, or were a sexually exploited minor, to three or more people in last six months.

### Injection drug use and history of stigma, discrimination, and criminalization

**Livingstone.** Median age when PWID in Livingstone started injecting was 19.0 years (IQR: 17.0–22.0) and they had injected for a median duration of 4.0 years (IQR: 2.0–6.0) (Table 3). Almost half (46.5%, 95% CI: 38.5–54.5) had injected drugs within a day of the survey, 30.1% (95% CI: 22.5, 38.0) reported injecting drugs two or more times per day in the last 6 months, 38.2% (95% CI: 30.7, 45.7) injected with a used needle at least once in the last 6 months, and 21.7% (95% CI: 14.6, 28.7) had engaged in Bluetooth (i.e., when an individual injects the blood of another person who had just injected a drug in order to share the high) at least once in their life. For a question that participants could select more than one response to, 92.5% (95% CI: 87.1, 98.8) reported that the most common drug they injected in the 6 months preceding the survey was Tie White Heroin, whereas a lower percentage cited Promethazine (8.8%, 95% CI: 2.7, 15.1), Artane (Trihexyphenidyl) (5.0%, 95% CI: 1.7, 8.3), and Benylin/Codein (3.9%, 95% CI: 0.6, 7.3). Due to injecting drugs, over half had been rejected by family (50.5%, 95% CI: 42.8, 58.2), 8.0% (95% CI: 3.8–12.3) reported being treated unfairly or denied care by healthcare practitioners (HCPs), 31.0% (95% CI: 23.6–38.4) avoided healthcare due to fear of someone discovering they inject, 20.2% (95% CI: 13.4, 26.7) had lost a job, and 14.4% (95% CI: 8.1, 20.5) have been arrested. Few were aware of drug treatment programs (including outpatient, inpatient, residential, detox, and methadone maintenance programs) (15.9%, 95% CI: 10.9, 20.8).

**Lusaka.** In Lusaka, the median age when PWID started injecting was 19.0 years (IQR: 17.0–23.0) and they had injected for a median duration of 5.0 years (IQR: 3.0–7.0). Almost half (44.6%, 95% CI: 37.6, 51.6) had injected within a day of the survey, 29.3% (95% CI: 23.0, 35.5) injected drugs two or more times per day in the last 6 months, 13.5% (95% CI: 9.0, 18.1) injected with a used needle at least once in the last 6 months, and 7.2% (95% CI: 2.6, 11.7) engaged in Bluetooth. Tie White Heroin was again the most commonly reported drug injected in the previous 6 months (95.8%, 95% CI: 93.5, 98.2), with a lower percentage reporting Mixed Heroin (76.4%, 95% CI: 70.2, 82.5), Cocaine (19.8%, 95% CI: 13.7, 25.9), and Promethazine (5.8%, 95% CI: 2.4, 9.3) as their injectable drug of choice. About two-thirds had been rejected by family (67.1%, 95% CI: 60.9, 73.4), 20.2% (95% CI: 14.8, 25.5) had been treated unfairly/denied care by HCPs, nearly half avoided healthcare (45.2%, 95% CI: 37.9, 52.4), a quarter lost a job (25.7%, 95% CI: 19.3, 32.1), and 37.6% (95% CI: 31.5, 43.9) had been arrested because they injected drugs. Approximately four in ten were aware of drug treatment programs (41.3%, 95% CI: 34.5, 48.2).

**Ndola.** The median age PWID in Ndola started injecting was 21.0 years (IQR: 18.0–25.0) and they had injected for a median of 6.0 years (IQR: 3.0–10.0). Almost a fifth (16.6%, 95% CI: 11.1, 22.0) had injected drugs within a day of the survey, 5.9% (95% CI: 2.6, 9.1) had injected two or more times a day for the last six months, 47.4% (95% CI: 39.3, 55.5)

**Table 3. Injection drug use and history of stigma, discrimination, and criminalization—Livingstone, Lusaka, and Ndola, Zambia, 2021.**

| | Livingstone | | | Lusaka | | | Ndola | | |
|---|---|---|---|---|---|---|---|---|---|
| **Sample Size** | **n = 235** | | | **n = 349** | | | **n = 259** | | |
| | **Sample** | | **Population** | **Sample** | | **Population** | **Sample** | | **Population** |
| | **n** | **%** | **% (95% CI)** | **n** | **%** | **% (95% CI)** | **n** | **%** | **% (95% CI)** |
| **Median age when started injecting drugs (IQR)** | 19.0 (17.0, 22.0) | | | 19.0 (17.0, 23.0) | | | 21.0 (18.0, 25.0) | | |
| **Age category when started injecting drugs** | | | | | | | | | |
| <15 | 19 | 8.1 | 5.7 (3.0-8.4) | 12 | 3.4 | 2.2 (0.8-3.7) | 7 | 2.7 | 2.3 (0.6-4.0) |
| 15-19 | 116 | 49.4 | 50.0 (41.0-59.0) | 172 | 49.4 | 51.8 (44.5-59.1) | 101 | 39.0 | 36.5 (28.4-44.5) |
| 20-24 | 52 | 22.1 | 24.8 (17.2-32.3) | 104 | 29.9 | 27.7 (21.7-33.7) | 82 | 31.7 | 34.4 (27.0-41.7) |
| ≥ 25 | 48 | 20.4 | 19.5 (13.5-25.5) | 60 | 17.2 | 18.3 (12.6-23.9) | 69 | 26.6 | 26.8 (19.4-34.2) |
| **Median duration of injection drug use (IQR), years** | 4.0 (2.0, 6.0) | | | 5.0 (3.0, 7.0) | | | 6.0 (3.0, 10.0) | | |
| **Last time injected drugs?** | | | | | | | | | |
| Today/yesterday | 122 | 51.9 | 46.5 (38.5-54.5) | 184 | 52.7 | 44.6 (37.6-51.6) | 47 | 18.2 | 16.6 (11.1-22.0) |
| Within last week | 92 | 39.1 | 43.3 (35.5-51.2) | 142 | 40.7 | 49.3 (41.7-56.8) | 125 | 48.4 | 46.8 (38.1-55.5) |
| Within last month or longer | 21 | 8.9 | 10.2 (5.7-14.5) | 23 | 6.6 | 6.1 (2.1-10.2) | 86 | 33.3 | 36.6 (27.5-45.8) |
| **How often were drugs injected in the last 6 months?** | | | | | | | | | |
| 2 or more times per day | 79 | 33.6 | 30.1 (22.5-38.0) | 120 | 34.4 | 29.3 (23.0-35.5) | 21 | 8.1 | 5.9 (2.6-9.1) |
| Once per day or less | 126 | 53.6 | 55.5 (47.2-63.4) | 135 | 38.7 | 36.4 (29.9-42.9) | 62 | 23.9 | 18.3 (12.4-24.1) |
| Once per week or less | 30 | 12.8 | 14.4 (9.3-19.6) | 94 | 26.9 | 34.3 (26.9-41.8) | 176 | 68.0 | 75.8 (69.0-82.8) |
| **Most common drugs injected in the last 6 months?** | | | | | | | | | |
| Tie White (Heroin) | 222 | 94.5 | 92.5 (87.1-98.8) | 333 | 95.4 | 95.8 (93.5-98.2) | 75 | 29.0 | 28.3 (19.9-36.6) |
| Dirty Drug/Voloo (Mixed Heroin) | 2 | 0.9 | 0.4 (0.0-0.8) | 263 | 75.4 | 76.4 (70.2-82.5) | 19 | 7.3 | 6.4 (2.3-10.5) |
| Artane (Trihexyphenidyl) | 10 | 4.3 | 5.0 (1.7-8.3) | 18 | 5.2 | 4.5 (1.9-7.2) | 177 | 68.3 | 68.2 (59.6-76.7) |
| Blue Marsh (Promethazine) | 15 | 6.4 | 8.8 (2.7-15.1) | 20 | 5.7 | 5.8 (2.4-9.3) | 160 | 61.8 | 60.1 (50.9-69.2) |
| Nyerere (Benylin/Codein) | 7 | 3.0 | 3.9 (0.6-7.3) | 22 | 6.3 | 4.3 (2.1-6.4) | 92 | 37.0 | 38.1 (30.4-46.0) |
| Ashtone Powder (Cocaine) | 1 | 0.4 | 0.4 (0.0-0.9) | 64 | 18.3 | 19.8 (13.7-25.9) | 29 | 11.2 | 11.7 (5.9-17.4) |
| **How often were drugs injected with a used needle in the last 6 months?** | | | | | | | | | |
| Never | 73 | 31.1 | 26.4 (20.5, 32.5) | 122 | 35.0 | 35.7 (28.8, 42.7) | 54 | 20.8 | 23.9 (16.5, 31.3) |
| At least once | 94 | 40.0 | 38.2 (30.7, 45.7) | 48 | 13.8 | 13.5 (9.0, 18.1) | 126 | 48.6 | 47.4 (39.3, 55.5) |
| No answer | 68 | 28.9 | 35.4 (27.1, 43.5) | 179 | 51.3 | 50.8 (43.9, 57.5) | 79 | 30.5 | 28.7 (21.7, 35.8) |
| **Ever engaged in Bluetooth?** | 48 | 20.4 | 21.7 (14.6-28.7) | 17 | 4.9 | 7.2 (2.6-11.7) | 32 | 12.4 | 10.5 (6.2-14.8) |
| **Rejected by family for injecting drugs** | 138 | 58.7 | 50.5 (42.8-58.2) | 234 | 67.0 | 67.1 (60.9-73.4) | 78 | 30.1 | 31.7 (23.9-39.5) |
| **Treated unfairly/denied care by HCPs because you inject drugs** | 19 | 8.1 | 8.0 (3.8-12.3) | 67 | 19.2 | 20.2 (14.8-25.5) | 43 | 16.6 | 18.6 (12.5-24.7) |
| **Avoided healthcare because of fear someone might discover they inject drugs** | 80 | 34.0 | 31.0 (23.6-38.4) | 159 | 45.6 | 45.2 (37.9-52.4) | 137 | 52.9 | 52.8 (44.5-61.2) |
| **Lost a job due to injecting drugs** | 44 | 18.7 | 20.2 (13.4-26.7) | 84 | 24.1 | 25.7 (19.3-32.1) | 36 | 13.9 | 16.9 (11.0-22.7) |
| **Ever arrested for injecting drugs** | 28 | 11.9 | 14.4 (8.1-20.5) | 133 | 38.1 | 37.6 (31.5-43.9) | 28 | 10.8 | 11.5 (6.5-16.5) |
| **Aware of any drug treatment programs*** | 42 | 17.9 | 15.9 (10.9-20.8) | 145 | 41.5 | 41.3 (34.5-48.2) | 51 | 19.7 | 20.0 (13.2-26.7) |

Population estimates and 95% Confidence Intervals are weighted using Gile's SS estimator, whereas sample estimates are unweighted. CI: Confidence Interval. IQR: Interquartile Range. *Includes outpatient, inpatient, residential, detox, and methadone maintenance drug treatment programs.

injected with a used needle at least once in the last 6 months, and 10.5% (95% CI: 6.2, 14.8) had engaged in Bluetooth. Artane (Trihexyphenidyl) was the most commonly injected drug (68.2%, 95% CI: 59.6, 76.7) in the previous 6 months, followed by Promethazine (60.1%, 95% CI: 50.9, 69.2), Benylin/Codein (38.1%, 95% CI: 30.4, 46.0), and Tie White Heroin (28.3%, 95% CI: 19.9, 36.6). Nearly a third had been rejected by family (31.7%, 95% CI: 23.9, 39.5), 18.6% (95% CI: 12.5, 24.7) had been treated unfairly/denied care by HCPs, 52.8% (95% CI: 44.5, 61.2) avoided healthcare, 16.9% (95% CI: 11.0, 22.7) lost a job, and 11.5% (95% CI: 6.5, 16.5) had been arrested because of injecting drugs. A fifth were aware of drug treatment programs (20.0%, 95% CI: 13.2, 26.7).

### HIV prevalence, viral load suppression, testing, outreach, and prevention awareness

**Livingstone.**  HIV prevalence among PWID in Livingstone was 11.9% (95% CI: 7.3, 16.5) (Table 4). Among PWID living with HIV, 70.8% (95% CI: 56.8, 85.1) were virally suppressed (< 1000 copies/mL). Of all PWID, 3.0% (95% CI: 0.6, 5.3) had active syphilis, and most had previously tested for HIV (91.8%, 95% CI: 87.4, 96.2). However, among those, 26.0% (95% CI: 20.0, 32.1) most recently tested more than a year before the survey. Almost two thirds (63.3%, 95% CI: 54.9, 71.5) received HIV information from a peer educator and/or outreach worker in their lifetime. For a question that participants could select more than one response to, 78.7% (95% CI: 71.9, 85.7) preferred getting HIV information from HCPs, whereas a lower percentage cited a preference for getting their information from peer educators and/or outreach workers (47.2%, 95% CI: 39.3, 55.0), friends (39.3%, 95% CI: 31.3, 47.3), internet (11.9%, 95% CI: 6.5, 17.3), television (8.5%, 95% CI: 4.5, 12.4), and/or radio (2.9%, 95% CI: 0.6, 5.3). Almost two-thirds of PWID (63.9%, 95% CI: 55.0, 72.9) heard of pre-exposure prophylaxis (PrEP). Of those, 16.6% (95% CI: 9.6, 23.8) had taken PrEP. Of those who had taken PrEP, 48.4% (95% CI: 32.9, 64.7) were on it at the time of the survey. The majority of PWID (93.6%, 95% CI: 88.3, 98.8) knew they were at risk for acquiring HIV by injecting with a used needle. The number of Hepatitis B and C cases among PWID was too low to report for Livingstone (≤5) and so were suppressed in Table 4 for confidentiality purposes.

**Lusaka.**  HIV prevalence among PWID in Lusaka was 7.3% (95% CI: 4.5, 10.2), of which 33.1% (95% CI: 17.7, 48.4) were virally suppressed. Of all PWID, 4.5% (95% CI: 2.1, 7.0) had active syphilis, 79.5% (95% CI: 74.2, 84.9) had tested for HIV; but of those, 36.5% (95% CI: 30.3, 42.7) had tested over a year before the survey. Of those who never tested, 23.6% (95% CI: 4.6, 42.6) did not test because they did not feel at risk whereas 22.8% (95% CI: 7.2, 38.2) did not test because they were afraid of receiving a positive result. The majority (90.0%, 95% CI: 86.5, 93.2) preferred getting HIV information from HCPs, whereas a lower percentage preferred the radio (56.6%, 95% CI: 49.6, 63.5), peer educators/ outreach workers (47.1%, 95% CI: 40.3, 54.0), friends (37.3%, 95% CI: 30.2, 44.4), and/or television (35.9%, 95% CI: 28.8, 42.9). Of all PWID, 16.0% (95% CI: 11.3, 20.8) heard of PrEP, and 15.8% (95% CI: 5.2, 26.3) of those had taken it. Of those who had taken PrEP, 71.2% (95% CI: 45.4, 98.2) were on it at the time of the survey. Lastly, 94.9% (95% CI: 92.0, 97.7) were aware injecting with used needles posed an HIV risk. Prevalence of Hepatitis B was 5.0% (95% CI: 1.7, 8.2) and the prevalence of Hepatitis C was too low to report (≤5).

**Ndola.**  HIV prevalence among PWID in Ndola was 21.9% (95% CI: 14.5, 29.3) and 58.0% (95% CI: 27.8, 88.3) of those were virally suppressed. Active syphilis prevalence was 11.1% (95% CI: 5.3, 16.8) and 49.3% (95% CI: 24.5, 75.3) of those were co-infected with HIV. The majority of PWID (87.0%, 95% CI: 81.0, 93.1) had been tested for HIV, but 32.4% (95% CI: 25.2, 39.6) of those tested more than a year before the survey. Among PWID who had never been tested, 36.8% (95% CI: 12.5, 62.0) did not test because they did not feel at risk and 29.9% (95% CI: 9.9, 50.0) did not test because they feared a positive result. Over half of all PWID (58.5%, 95% CI: 50.6, 66.4) had received HIV counseling from a peer educator or outreach worker. Most (84.1%, 95% CI: 78.3, 90.1) stated a preference for receiving HIV information from HCPs. Please see Table 4 for percentages who preferred other sources. Over half (53.3%, 95% CI: 43.9, 62.8) heard of PrEP and 23.0% (95% CI: 8.9, 37.0) of those had taken it. Of those who had taken PrEP, 70.3% (95% CI: 53.2, 87.1) were on it at the time of the survey. Most PWID (96.7%, 95% CI: 94.5, 98.9) knew they could get HIV from used needles. Hepatitis B prevalence was 2.2% (95% CI: 0.7, 3.7) and Hepatitis C prevalence was too low to report (n ≤ 5).

**Table 4. HIV prevalence, viral load suppression, biomarker testing, outreach, PrEP, and prevention awareness—Livingstone, Lusaka, and Ndola, Zambia, 2021.**

| | Livingstone | | | Lusaka | | | Ndola | | |
|---|---|---|---|---|---|---|---|---|---|
| **Sample Size** | n = 235 | | | n = 349 | | | n = 259 | | |
| | **Sample** | | **Population** | **Sample** | | **Population** | **Sample** | | **Population** |
| | n | % | % (95% CI) | n | % | % (95% CI) | n | % | % (95% CI) |
| **HIV Prevalence** | 32 | 13.6 | 11.9 (7.3-16.5) | 26 | 7.4 | 7.3 (4.5-10.2) | 52 | 20.1 | 21.9 (14.5-29.3) |
| **Viral Load Suppression** [1] (of those who are HIV-positive) | 23 | 71.9 | 70.8 (56.8-85.1) | 9 | 34.6 | 33.1 (17.7-48.4) | 29 | 55.8 | 58.0 (27.8-88.3) |
| **Active Syphilis Prevalence** | 9 | 3.8 | 3.0 (0.6-5.3) | 16 | 4.6 | 4.5 (2.1-7.0) | 26 | 10.0 | 11.1 (5.3-16.8) |
| **HIV/Active Syphilis Co-infection** (of those with active syphilis) | * | * | * | * | * | * | 15 | 57.7 | 49.3 (24.5-75.3) |
| **Previously tested for HIV** | 215 | 91.5 | 91.8 (87.4-96.2) | 282 | 80.8 | 79.5 (74.2-84.9) | 228 | 88.0 | 87.0 (81.0-93.1) |
| **Never tested for HIV** | 20 | 8.5 | 8.2 (3.8-12.5) | 67 | 19.2 | 20.5 (15.2-25.7) | 31 | 12.0 | 13.0 (7.5-18.4) |
| Did not test because they did not feel at risk | * | * | * | 16 | 23.9 | 23.6 (4.6-42.6) | 10 | 32.3 | 36.8 (12.5-62.0) |
| Did not test because they feared a positive result | * | * | * | 14 | 20.9 | 22.8 (7.2-38.2) | 9 | 29.0 | 29.9 (9.9-50.0) |
| **Timing of last HIV test** | | | | | | | | | |
| In the last 6 months | 111 | 47.2 | 51.0 (43.0-59.0) | 108 | 30.9 | 25.5 (20.2-31.0) | 105 | 40.5 | 44.2 (35.7-52.6) |
| Between 7–12 months | 32 | 13.6 | 14.8 (9.5-20.1) | 49 | 14.0 | 16.8 (11.2-22.3) | 35 | 13.5 | 10.3 (5.4-15.2) |
| More than 12 months | 72 | 30.6 | 26.0 (20.0-32.1) | 124 | 35.5 | 36.5 (30.3-42.7) | 87 | 33.6 | 32.4 (25.2-39.6) |
| **Received HIV messaging from peer educator or outreach worker in their lifetime** | 147 | 62.6 | 63.3 (54.9-71.5) | 196 | 56.2 | 55.9 (48.6-63.3) | 152 | 58.7 | 58.5 (50.6-66.4) |
| **Preferred source for HIV information** | | | | | | | | | |
| Health care providers | 188 | 80.0 | 78.7 (71.9-85.7) | 297 | 85.1 | 90.0 (86.5-93.2) | 214 | 82.6 | 84.1 (78.3-90.1) |
| Peer educator/outreach worker | 101 | 43.0 | 47.2 (39.3-55.0) | 162 | 46.4 | 47.1 (40.3-54.0) | 76 | 29.3 | 31.0 (22.7-39.3) |
| Internet | 24 | 10.2 | 11.9 (6.5-17.3) | 32 | 9.2 | 9.1 (5.1-13.0) | 13 | 5.0 | 5.7 (1.6-9.8) |
| Television | 27 | 11.5 | 8.5 (4.5-12.4) | 118 | 33.8 | 35.9 (28.8-42.9) | 19 | 7.3 | 5.0 (2.5-6.5) |
| Radio | 6 | 2.6 | 2.9 (0.6-5.3) | 188 | 53.9 | 56.6 (49.6-63.5) | 18 | 6.9 | 6.5 (2.8-10.2) |
| Friends | 75 | 31.9 | 39.3 (31.3-47.3) | 118 | 33.8 | 37.3 (30.2-44.4) | 30 | 11.6 | 13.8 (8.1-19.4) |
| **Ever heard of PrEP** | 152 | 67.6 | 63.9 (55.0-72.9) | 63 | 19.0 | 16.0 (11.3-20.8) | 116 | 49.4 | 53.3 (43.9-62.8) |
| **Taken PrEP** (of those who have heard of it) | 24 | 15.8 | 16.6 (9.6-23.8) | 10 | 15.9 | 15.8 (5.2-26.3) | 29 | 25.0 | 23.0 (8.9-37.0) |
| **Currently on PrEP** (of those who have taken it) | 11 | 45.8 | 48.4 (32.9-64.7) | 6 | 60.0 | 71.2 (45.4-98.2) | 22 | 75.9 | 70.3 (53.2-87.1) |
| **Aware can get HIV by injecting with used needle** | 219 | 93.2 | 93.6 (88.3-98.8) | 332 | 95.1 | 94.9 (92.0-97.7) | 250 | 96.5 | 96.7 (94.5-98.9) |
| **Hepatitis B (HBsAg) Prevalence** | * | * | * | 16 | 4.6 | 5.0 (1.7-8.2) | 10 | 3.9 | 2.2 (0.7-3.7) |
| **Hepatitis C prevalence‡** | * | * | * | * | * | * | * | * | * |

Population estimates and 95% Confidence Intervals are weighted using Gile's SS estimator, whereas sample estimates are unweighted. CI: Confidence Interval. IQR: Interquartile Range. HBsAg: Hepatitis B surface antigen. *Suppressed due to number being ≤ 5. [1] Viral Load Suppression (VLS) is < 1000 copies/mL. ‡There were five HCV cases across all three sites.

## Population-level 95-95-95 progress

**Livingstone.** Of all HIV-positive PWID in Livingstone, 70.7% (95% CI: 55.4, 85.0) were aware of their serostatus. Of those, 100% were on ART, and 100% of those had achieved VLS (Fig 1).

**Lusaka.** Among HIV-positive PWID in Lusaka, 66.0% (95% CI: 49.3, 82.2) were aware of their serostatus. Of those, 75.7% (95% CI: 51.1, 99.9) were on ART and 66.3% (95% CI: 42.1, 90.9) of those had achieved VLS.

**Ndola.** Of the HIV-positive PWID in Ndola, 60.2% (95% CI: 44.1, 76.0) were aware of their serostatus, 100% of those were on ART, and 90.2% (95% CI: 82.2, 98.3) of those on ART had achieved VLS.

## Discussion

Our results summarize the first probability-based BBS among PWID in Zambia and include information on HIV, VL, syphilis, HBV, HCV, and progress towards the UNAIDS 95-95-95 targets. Also presented are key sociodemographic and sexual history patterns, along with findings on injection drug use, stigma, discrimination, and criminalization that show-case commonalities and variations across sites. Livingstone's high HIV prevalence, low HIV risk perception, young PWID population, higher prevalence of forced sex, and lower awareness of drug treatment programs present unique challenges. Lusaka has lower HIV prevalence but a significant gap in ART uptake and VLS, very low awareness of PrEP, and demonstrates a high proportion of male PWID, homelessness, and a substantial number forced to have sex. Ndola's slightly older PWID population includes a much larger female presence and larger prevalence of those who inject with used needles, which may explain its especially high HIV prevalence and subsequent need for expanded preventive interventions. The site-specific differences revealed by this survey contribute to the broader discourse on HIV among PWID in SSA and emphasize the importance of community-specific approaches in addressing the diverse challenges faced by PWID in different contexts.

### Livingstone

HIV prevalence among PWID in Livingstone (11.9%, 95% CI: 7.3, 16.5) was consistent with HIV prevalence among those 15 years and older in the general population of Zambia's Southern province (13.2%), where Livingstone is located [20]. However, it is higher than the prevalence in 20–24-year-old women (5.9%) and men (1.8%) of the country-wide general population, which is roughly the same age range as our sample [20]. CIs were wide so results should be interpreted with caution, but findings indicate the 2nd and 3rd 95 targets were met among PWID in Livingstone. However, their progress toward the 1st

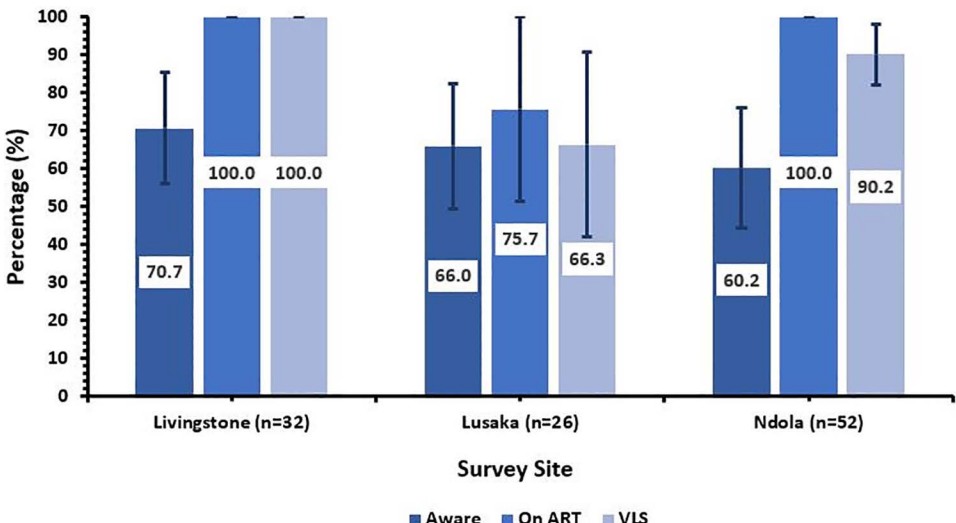

**Fig 1. Progress towards UNAIDS 95-95-95 targets among PWID living with HIV in Livingstone, Lusaka, and Ndola, Zambia, 2021.** Population estimates and 95% Confidence Intervals are weighted using Gile's SS estimator. ART status is viral load adjusted (<200 copies/mL). VLS is defined as <1000 copies/mL. Conditional percentages are shown.

95 was well below the target. Several issues could have contributed to this observation. Almost a third of Livingstone's PWID who had previously tested for HIV had tested more than a year prior to the survey. WHO guidelines state that PWID should be tested every year [48], whereas Zambian guidelines state that persons should be retested after instances of having shared injecting equipment [33] and almost four in ten PWID in Livingstone injected with used needles at least once in the six months preceding the survey. Stigma could have played a role in this community delaying or avoiding testing as well, since almost one-third said they avoided healthcare due to fear someone might discover they inject drugs.

### Lusaka

PWID in Lusaka had an HIV prevalence (7.3%, 95% CI: 4.5, 10.2) slightly higher than women (5.9%) and men (1.8%) of Zambia's general population of the same age group but lower than the all-age prevalence in the broader Lusaka province's general population (14.4%) [20]. It may have been lower than the general population all-age prevalence due to a combination of the young age and preponderance of males (97.4%, 95% CI: 96.0, 98.9) in Lusaka's PWID population. Compared to the other two cities, Lusaka showed relatively poor progress toward all three 95 targets. It had the lowest percentage of PWID who had ever been tested for HIV, and of those, over a third were not tested within the previous year. Compared to Livingstone (31.0%, 95% CI: 23.6, 38.4), a higher proportion of PWID in Lusaka (45.2%, 95% CI: 37.9, 52.4) avoided healthcare due to fear of someone discovering they inject drugs, and almost 8.0 percentage points less of PWID had been reached by peer educators and/or outreach workers. About 40% had been arrested for injection drug use, which is part of an overall shift within Zambia's capital toward punitive, rather than rehabilitative drug enforcement policies, particularly among young people [49]. One-fifth of Lusaka PWID had been treated unfairly and/or outright denied care by HCPs due to injection drug use. Their relatively low rate of HIV testing compared to Livingstone and Ndola, low rate of recent testing, high frequency of healthcare avoidance, unfair treatment or denial of care by HCPs, low number of outreach encounters, and high arrest rates all could have contributed to Lusaka PWID's lower performance toward the UNAIDS goals.

### Ndola

HIV prevalence among PWID in Ndola (21.9%, 95% CI: 14.5, 29.3) was higher than that of the general all-age population in the Copperbelt province (11.9%) [20], where Ndola is located. It was also higher than women (12.0%) and men (3.1%) in the 25–29-year-old age group of the Zambian general population [20]. The median age of PWID from Ndola was 28.0 years. Prevalence of active syphilis in Ndola (11.1%) was more than double the prevalence of the other two cities, and of those with active syphilis, nearly half were HIV-positive. Progress toward the 1st 95 was low (60.2%). Like Lusaka, about a third of PWID who had ever been tested for HIV had last been tested more than a year before the survey. Risk perception was even lower in Ndola than the other two sites: almost 40% of those never tested for HIV said their reason for not testing was because they felt they were not at risk and almost half of PWID in Ndola admitted to injecting with a used needle at least once in the 6 months preceding the survey despite almost all knowing they can get HIV by injecting with a used needle. Once aware of their status, PWID in Ndola seemingly did not have a problem initiating ART, but only 90.2% of those on ART achieved VLS, which indicates that adherence to ART or the prescribing of proper ART regimens and follow-up may be a problem. Women comprised nearly 30% of the sample in Ndola compared to over 15% in Livingstone and just over 2% in Lusaka, which likely contributed to the high prevalence of active syphilis observed. Female PWID typically experience higher rates of sexually transmitted infections than male PWID [50]. It would be advisable for future inquiries to ascertain if there are reasons other than the higher share of female PWID as to why HIV and active syphilis prevalence is so high in Ndola.

### Implications for programs and policy

While there were similarities across the three sites, there were notable differences as well. This granularity in results would not have been detected with a national-level survey, provincial-level survey, or if our site-specific data had been aggregated.

The community-level data obtained from this first ever BBS among PWID in Livingstone, Lusaka, and Ndola establish an understanding of where specific gaps exist regarding 95-95-95 progress and provide insights into what may be contributing to these gaps. Local data like this can help stakeholders better design and implement context-driven interventions meant to maximize the number of PWID who are aware of their HIV serostatus, on ART, and who have achieved VLS.

All three sites were well below the 1st 95 target regarding serostatus awareness. A positive finding was that over 90% of all PWID across sites were aware HIV could be transmitted by sharing needles. A 2015 study in Mwanza, Tanzania showed that only 64.1% of PWID knew sharing needles posed an HIV risk [51]. The high level of awareness regarding the harms of needle sharing contrasts with the appreciable share of PWID in Livingstone and Ndola who reported injected with used needles. It also contrasts with a large portion of PWID in Lusaka and Ndola who had never tested for HIV due to feeling they were not at risk, and that roughly a third of those who had tested for HIV did so more than a year before the survey. Also, approximately only a fifth of PWID across all sites who were aware of PrEP had ever taken it. In Livingstone, Lusaka, and Ndola the preferred source of HIV information was HCPs. The aforementioned findings highlight a potential opportunity for HCPs to review national guidelines and update them as necessary to meet the unique gaps and needs of PWID. Additional training for HCPs that help them initiate discussions with patients on why they are not testing more often and/or why they do not feel at risk may be helpful. Other potential reasons for lack of testing and lack of recent testing are the commonly reported issues of avoiding healthcare in the first place due to fear of injection drug use discovery and that PWID who do visit HCPs may not be disclosing their drug use, making it impossible for providers to give injection drug use-specific counseling. Local health departments could ensure the existence or expansion of in-service training for HCPs that includes a robust curriculum on the most up-to-date HIV treatment, prevention, and counseling guidelines, sensitization training on key populations and harm reduction, along with improved outreach efforts as well [48]. In addition, only about 60% of all PWID had been counseled by peer educators and/or outreach workers about HIV awareness and prevention. Better counseling by HCPs and an increased number of quality encounters with HIV outreach workers and peer educators may increase HIV risk perception, knowledge profiles, and outcomes [52–54]. It may be needed for peer educators and outreach workers to be more proactive about creating new connections with PWID reluctant to seek out services on their own. HCPs, peer educators, and outreach workers should strive to work together to educate PWID about HIV, whereas HCPs and case managers may be able to navigate them more effectively through the HIV prevention and care continuum. To decrease the concerns PWID have of their drug use being discovered, partnerships can be created between interest holders (e.g., PEPFAR, Global Fund) and the Zambian MoH to increase the number of PWID friendly service providers and community-based HIV facilities (i.e., safe environments with decreased threat of stigma, discrimination, or having injection drug use reported to the police [12,55,56]) where positive support and education networks among peers, HCPs, and outreach personnel can be fostered [52,57–59]. Preventive resources like PrEP can be efficiently offered in these facilities as well [59], along with OAMT [60] and NSPs. Results from the 2021 BBS have already propelled the establishment of a PWID task force with representation from government and non-governmental agencies that is working to establish MAT clinics while the police and Home Affairs Ministry offer basic rehabilitative services to the public in Lusaka. The PWID task force is also in the process of creating national policy regarding methadone approval in Zambia. In preparation for its approval, University Teaching Hospital in Lusaka is preparing space for a MAT clinic, but at the current time is only offering services like HIV testing, treatment, and wound care to attract PWID clientele for when OAMT becomes approved. Again, OAMT and NSPs, if approved, would likely serve as a strong facilitator to getting PWID into the HIV testing and treatment pathway. To this end, OAMT may be most impactful in Livingstone and Lusaka where Heroin is by far the most commonly reported drug injected, as opposed to Ndola where Trihexyphenidyl is most common; whereas NSPs may be most impactful in Livingstone and Ndola, where PWID reported higher rates of injecting with a used needle than Lusaka.

The low HBV and HCV prevalence across all three sites is encouraging. Active HBV/HCV surveillance and testing should continue within the high-risk PWID community, while consideration can be given to streamlined treatment models, such as point-of-service care [61].

## Limitations

Independent associations between plausible predictive factors and the 95 target outcomes (e.g., prevalence of HIV awareness, on ART, and VLS) were explored. Unfortunately, sample sizes were very small for certain indicators which resulted in CIs and standard errors too large to convey meaningful information. Therefore, these results were not reported. Most PWID said they preferred to receive HIV information from HCPs, but more detailed data on those interpersonal interactions were not available, such as whether HCPs created a PWID-friendly environment or not, how comfortable they made it for PWID to disclose injection drug use status, what kind of proactive outreach they may have done to contact PWID, what type of information HCPs typically gave to PWID after becoming aware of their HIV status, problems that may have occurred in prescribing the full correct ART regimen, and/or other barriers such as distance to health facilities. Our variable on awareness of drug treatment programs (including outpatient, inpatient, residential, detox, and methadone maintenance therapy) is of limited value since OAMT and NSPs are currently not available in Zambia, however the low awareness observed indicates work must be done to increase awareness on the benefits of these programs if they are to be maximally successful if and when they are approved. Data on issues like HIV drug resistance, stockouts, distance to facilities, guidance on same day ART initiation, and other facilitators and barriers that could have impacted progress toward the 95-95-95 targets were not available. This survey also took place during the COVID Omicron wave which could have impacted network size, weighting, and access to HIV-related services [62]. Lastly, this is a cross-sectional survey that is subject to recall bias [63] and social desirability bias [64], especially since many questions were regarding sensitive issues about injection drug use and sexual and reproductive health history.

## Conclusion

This survey produced a representative sample of PWID in Livingstone, Lusaka, and Ndola, Zambia and is a foundation for understanding HIV vulnerabilities in this population. It demonstrated that PWID in all three sites were well short of achieving the 1st UNAIDS 95 target. Achievement toward the 2nd and 3rd targets were variable across sites. Only about 60% of PWID had been in contact with peer educators and/or outreach workers, awareness of drug treatment programs was low, needle sharing was common, many PWID who had been tested for HIV did so more than a year before the survey, and an appreciable number of respondents avoided healthcare due to fear of someone discovering their injection drug use. Improved counseling by HCPs; the establishment of community-based PWID-friendly HIV, OAMT, and NSP clinics that can serve as another link for helping PWID access HIV testing; and better coordination between HCPs, peer educators, and outreach workers in proactively contacting, counseling, and guiding PWID through the HIV prevention and care continuum may help improve these outcomes.

The geographical nuances to barriers and characteristics of HIV prevention, awareness, and care uncovered in this survey contribute to the larger discussion on HIV in SSA and emphasize the importance of site-specific approaches to addressing the unique challenges faced by PWID. Insights gleaned from this survey can be used to justify increased support and funding for expanded HIV interventions among PWID in Zambia. This survey also indicates that strengthening PWID-specific programs can advance progress toward the overall UNAIDS targets and better ensure the well-being of this marginalized population.

## Acknowledgments

We extend our gratitude to all participants for dedicating their time and providing invaluable feedback. Our sincere appreciation goes to our partners, and we acknowledge the significant contributions of the Zambia Survey Advisory Group, which includes representatives from CDC, NAC, TDRC, ICAP, and the Zambia Key Population Consortium.

## Author contributions

**Conceptualization:** Anne F McIntyre, Hiwote Solomon, Brave Hanunka, Lazarus Chelu, Ray Handema, Shepherd Khondowe, Kelvin Kapungu, Innocent Bwalya, Chipili Mulemfwe, Joyce J Neal, Maria Lahuerta, Lauren E Parmley, John Mwale, Lloyd B Mulenga.

**Data curation:** Brave Hanunka, Lazarus Chelu, Tepa Nkumbula, Ray Handema, Shepherd Khondowe, Kelvin Kapungu, Innocent Bwalya, Chipili Mulemfwe, Giles Reid, Maria Lahuerta, Lauren E Parmley.

**Formal analysis:** Daniel Woytowich, Anne F McIntyre, Hiwote Solomon, Tepa Nkumbula, Giles Reid.

**Funding acquisition:** Maria Lahuerta, Lauren E Parmley.

**Investigation:** Anne F McIntyre.

**Methodology:** Daniel Woytowich, Anne F McIntyre, Brave Hanunka, Lazarus Chelu, Ray Handema, Shepherd Khondowe, Kelvin Kapungu, Innocent Bwalya, Chipili Mulemfwe, Kennedy Mutale, Giles Reid, Joyce J Neal, Maria Lahuerta, Lauren E Parmley, Hannah Chung, Avi J Hakim, John Mwale.

**Project administration:** Neena M Philip, Maria Lahuerta, Lloyd B Mulenga.

**Supervision:** Anne F McIntyre, Hiwote Solomon, Maria Lahuerta, Lauren E Parmley, Lloyd B Mulenga.

**Validation:** Daniel Woytowich, Anne F McIntyre, Hiwote Solomon, Kennedy Mutale, Neena M Philip.

**Visualization:** Daniel Woytowich.

**Writing – original draft:** Daniel Woytowich, Anne F McIntyre, Hiwote Solomon, Brave Hanunka.

**Writing – review & editing:** Daniel Woytowich, Anne F McIntyre, Hiwote Solomon, Brave Hanunka, Lazarus Chelu, Tepa Nkumbula, Leigh Tally, Ray Handema, Shepherd Khondowe, Kelvin Kapungu, Lophina Chilukutu, Innocent Bwalya, Chipili Mulemfwe, Melvin Mwansa, Kennedy Mutale, Neena M Philip, Joyce J Neal, Maria Lahuerta, Lauren E Parmley, Hannah Chung, Avi J Hakim, Jonas Z Hines, Evelyn Kim, John Mwale, Lloyd B Mulenga.

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
