## [Decision Letter · Decision Letter 0]

7 Oct 2024

PONE-D-24-29546Local data for local programming: Results from an HIV biobehavioral survey among people who inject drugs in Livingstone, Lusaka, and Ndola, Zambia, 2021PLOS ONE

Dear Dr. Woytowich,

Thank you for submitting your manuscript to PLOS ONE. After careful consideration, we feel that it has merit but does not fully meet PLOS ONE’s publication criteria as it currently stands. Therefore, we invite you to submit a revised version of the manuscript that addresses the points raised during the review process.

We look forward to receiving your revised manuscript.

Kind regards,

Andrew Scheibe, MBChB MPH

Academic Editor

PLOS ONE

Journal requirements: When submitting your revision, we need you to address these additional requirements. 1. Please ensure that your manuscript meets PLOS ONE's style requirements, including those for file naming. The PLOS ONE style templates can be found at https://journals.plos.org/plosone/s/file?id=wjVg/PLOSOne_formatting_sample_main_body.pdf and https://journals.plos.org/plosone/s/file?id=ba62/PLOSOne_formatting_sample_title_authors_affiliations.pdf" 2. Please include a complete copy of PLOS’ questionnaire on inclusivity in global research in your revised manuscript. Our policy for research in this area aims to improve transparency in the reporting of research performed outside of researchers’ own country or community. The policy applies to researchers who have travelled to a different country to conduct research, research with Indigenous populations or their lands, and research on cultural artefacts. The questionnaire can also be requested at the journal’s discretion for any other submissions, even if these conditions are not met.  Please find more information on the policy and a link to download a blank copy of the questionnaire here: https://journals.plos.org/plosone/s/best-practices-in-research-reporting. Please upload a completed version of your questionnaire as Supporting Information when you resubmit your manuscript. 3. We note that the grant information you provided in the ‘Funding Information’ and ‘Financial Disclosure’ sections do not match.  When you resubmit, please ensure that you provide the correct grant numbers for the awards you received for your study in the ‘Funding Information’ section. 4. Thank you for stating the following financial disclosure:  [This activity was supported by the US President’s Emergency Plan for AIDS Relief (PEPFAR) (https://www.state.gov/pepfar/) through the US Centers for Disease Control and Prevention (CDC) under the terms of cooperative agreement “Supporting sustainable surveillance systems among key populations and support the Government of Zambia to improve HIV-related services for KPs” (Prime Award No. 1NU2GGH002056). The findings and conclusions in this report are those of the author(s) and do not necessarily represent the official position of the funding agencies. ].  Please state what role the funders took in the study.  If the funders had no role, please state: ""The funders had no role in study design, data collection and analysis, decision to publish, or preparation of the manuscript."" If this statement is not correct you must amend it as needed. Please include this amended Role of Funder statement in your cover letter; we will change the online submission form on your behalf. 5. We note that you have indicated that there are restrictions to data sharing for this study. For studies involving human research participant data or other sensitive data, we encourage authors to share de-identified or anonymized data. However, when data cannot be publicly shared for ethical reasons, we allow authors to make their data sets available upon request. For information on unacceptable data access restrictions, please see http://journals.plos.org/plosone/s/data-availability#loc-unacceptable-data-access-restrictions.  Before we proceed with your manuscript, please address the following prompts: a) If there are ethical or legal restrictions on sharing a de-identified data set, please explain them in detail (e.g., data contain potentially identifying or sensitive patient information, data are owned by a third-party organization, etc.) and who has imposed them (e.g., a Research Ethics Committee or Institutional Review Board, etc.). Please also provide contact information for a data access committee, ethics committee, or other institutional body to which data requests may be sent. b) If there are no restrictions, please upload the minimal anonymized data set necessary to replicate your study findings to a stable, public repository and provide us with the relevant URLs, DOIs, or accession numbers. Please see http://www.bmj.com/content/340/bmj.c181.long for guidelines on how to de-identify and prepare clinical data for publication. For a list of recommended repositories, please see https://journals.plos.org/plosone/s/recommended-repositories. You also have the option of uploading the data as Supporting Information files, but we would recommend depositing data directly to a data repository if possible. Please update your Data Availability statement in the submission form accordingly.

Additional Editor Comments:

Thank you for submitting this paper. It is great to have information on PWID from Zambia in the literarture. In addition to the feedback provided by the reviews:

1. Introduction (lines 82 - 84). The objective does not mention assessment or presentation of results around syphylis, HBV or HCV (STI and BBV), so consider noting these too.

2. Introduction. The provision and availability of needle and syringe programmes (NSP) and opioid agonist maintenance therapy (OAMT) and treatment of stimulant dependence in Zambia should be briefly described.

3. Methods (lines 92). Insert a short description about each study site for readers who may not be familiar with Zambia.

4. Findings. The findings section (notably descriptions) is a bit long in parts. Aim to focus on key findings, and reduce some descriptions of socio demographic characteristics, and refer to the table.

5,.Findings. Table 1. Insert the % for all rows, where relevant (currently only done for some).

6. Findings. Unless I missed it, I did not see the description of the type of drugs injected. This is critical to assess risk, prevention and interventions.

7. Table 3. Median duration of injecting. Check the median and p25 and p75 - I would expect the median to lie between these, but does not seem to be the case (e.g., Ndola median is 3.0 years, but IQR is 6 - 10 year).

8. Insert n for anti-HCV results and HCV PCR for sample in table and findings description.

9. Findings around awareness of OAMT - this would only apply to people with opioid dependence, but it is not clear what drugs people injected. Also, Table 3 notes " aware of any drug treatment programs" - this is too broad to reflect insights into OAMT. So if questions on OAMT were asked, please include. If not, this could be noted in the limitations.

10. Findings. Why no findings of needle reuse/ sharing? This is the most important risk factor for PWID? (Unless I missed it). This is key to include, as it is a UNAIDS GAM indicator. Please include these variables.

11. Discussion. Among PWID, the primary HIV prevention interventions are NSP and OAMT, these should be foregrounded. These interventions are more important for PWID than PrEP, yet there is much emphasis on PrEP. Similarly, the implications for policy and practice, NSP and OAMT need to be much higher priorities than PrEP, but the framing does not align with emphasis on the issues that are particular to PWID. There is mention of MAT clinics, but what about NSP? This is critical, as many African countries are resistant to NSP, but without NSP HIV prevention interventions for PWID will fail to be be effective.

12. Discussion. The low HBV and (HCV?) prevalence is encouraging, and the implication for policy could also be to continue active surveillance and testing in these high risk groups to control the epidemic, and have access to HCV treatment early, before there is an epidemic, like is happening in other countries (e.g., South Africa).

13. Conclusion. Reflection on the important gap of NSPs seems to be missing. NSPs should be framed as being the link to access HTS and then enter the treatment pathway.

We look forward to receiving your revised manuscript.

Reviewers' comments:

Reviewer's Responses to Questions

**Comments to the Author**

1. Is the manuscript technically sound, and do the data support the conclusions?

Reviewer #1: Yes

Reviewer #2: Yes

2. Has the statistical analysis been performed appropriately and rigorously? 

Reviewer #1: N/A

Reviewer #2: Yes

3. Have the authors made all data underlying the findings in their manuscript fully available?

Reviewer #1: Yes

Reviewer #2: No

4. Is the manuscript presented in an intelligible fashion and written in standard English?

Reviewer #1: Yes

Reviewer #2: Yes

5. Review Comments to the Author

Reviewer #1: The details from this survey are nicely presented, and provide a rich, detailed biobehavioural insights into PWID risks in Zambia. I think the manuscript can however be too descriptive in parts, and some statistical analysis would complement the data nicely.

Abstract

Lines 38-39 – don’t need to mention both HIV and HIV viral load testing.

Results – find the (1st 95) confusing, same for 2nd and 3rd – link to targets unclear. For those less familiar with HIV targets, it may be helpful to say what the targets are when they’re being discussed (and elsewhere in the paper).

Manuscript

Line 96 – the ‘prevalence of VLS’ may be better described as the proportion of people living with HIV with VLS

Lines 102-103 – not sure the data on sample size calculations are needed if they were missed by such a large margin. If the authors want to include the calculation, more details are needed on the factors limiting study size (costs/ time constraints etc).

Lines – pre-test counselling for HIV/syphilis is mentioned, was this also done for HBV/HCV, or are there no national guidelines on this?

Line 142 – Might be useful to state where the Tropical Diseases Research Centre (TDRC) laboratory is based

Results lines 196-320, Tables 2-4 – this section contains a large amount of data in well-presented tables but overall, is very descriptive with the paragraphs describing the data from each table for each location. I would like to see more comparison between the sites in this section – ie were people younger/older at one site? Were there different risk factors at each site? Was one site better at getting PWID onto ART? Some of this is mentioned in the discussion but is perhaps better placed in the results section. Some statistics to highlight differences between the sites would be informative and help to summarise the data.

Discussion – it would be useful to get some idea of how widespread injection drug use is in Zambia, and particularly in Ndola where the HIV prevalence among PWID is very high.

Reviewer #2: General comment:

Are there any prevention services available for PWID specifically in Zambia, either NSP or OAMT? The instrument asked about OAMT awareness but were there questions about uptake? Or where needles were obtained?

In comparing the results in the article with the IBBS report found online the values do not match (e.g. HIV prevalence in Lusaka is 21.9% in the article vs 21.3% in the report; the 95’s for Ndola in the article are 60.2%-100%-90.2% while in the report there are 61.9%-100%-83.7%… ). Could these discrepancies be clarified for readers who may look at the report, too?

Line 34: “progress of PWID”…is it progress of the PWID themselves or the epidemic response for and by them? It reads awkwardly as written.

Line 59: “Whereas the world’s general population has an HIV prevalence of 0.7%…”

It would be worth to specify if this refers to all ages, 15+ or any other age range. The same applies to all the values related to the general population in the article (e.g. Line 61).

Also, the value would need bounds, kindly add them, as well as all the rest of the values in the article.

Line 63: “…in which 19.6% of SSA PWID participate.”

The source from where the data come from specify that the people have participated recently.

Line 68: “… defined as HIV RNA of <1000 copies/mL during the most recent VL test”.

The source for this is: 2016 World Health Organization (WHO) Consolidated guidelines on the use of antiretroviral drugs for treating and preventing HIV infection. Kindly update it.

Lines 96-97: “…people living with HIV in Zambia was 59.2% (19).”

The correct citation for this seems to be "Ministry of Health, Zambia. Zambia Population-based HIV Impact Assessment (ZAMPHIA) 2016: Final Report. Lusaka, Ministry of Health. February 2019".

Line 101: “…assuming an HIV prevalence of 25%”.

Why assume a prevalence of 25% if the evidence shown on the introduction showed 15.2% globally and 11.2% in SSA?

Line 236: “…healthcare because they inject drugs…”

Please specify that they have being treated unfairly or denied healthcare by healthcare practitioners.

Line 253: “…injected for a median duration of 3 years (IQR: 6-10).”

Why the IQR do not have decimal numbers? Please consolidate with the rest of the article.

Line 266: “…with HIV, 70.8% (95% CI: 56.8, 85.1) were virally suppressed”. & Line 294 “…and 58.0% (95% CI: 27.8, 88.3) of…”

Why to use the cascade here if in the abstract and introduction the reference is the 95-95-95?

Lines 273-274: “…from peer educators and/or outreach workers (47.2%), internet (11.9%), television (8.5%), radio (2.9%), and/or friends (31.9%).” & Lines 288-289 “…workers (47.1%), internet (9.1%), television (35.9%), radio (56.6%), and/or friends (37.3%).”& Lines 303-304 “…workers (31.0%), the internet (5.7%), television (5.0%), radio (6.5%), and/or friends (13.8%).”

Why not adding IC 95% on these values?

Line 277: “…Number of Hepatitis B cases…”.

Could you kindly add that Hepatitis C cases too?

Line 284: “… because they did not feel at risk whereas 22.8% (7.2, 38.2) did…”

Please add that numbers within brackets correspond to 95% CI.

Line 332: “…forced sex, and lower awareness of non-injectable alternatives present unique challenges.”

Please specify where this sentence comes from. It hasn't been introduced through the article.

Line 343: “…(13.2%), where Livingstone is located (19).” & Line 344 “…women (5.9%) and men (1.8%) of the country-wide…” & Lines 354-355-356 “…slightly higher than women (5.9%) and men (1.8%) of Zambia’s general population of the same age group but lower than the all-age prevalence in the broader Lusaka province’s general population (14.4%) (19)” & Lines 373-374: “…Copperbelt province (11.9%) (19), where Ndola is located. It was also higher than women (12.0%) and men (3.1%) in…”

Specially interesting to include IC 95% here to see if those overlap with the results found.

Line 362: “…discovering they inject drugs, and almost 8.0%…”

Is 8 percentual points, may be confusing, perhaps helpful to clarify.

Line 368: “… avoidance of healthcare,…”

Please include that it is “high frequency of” avoidance of healthcare.

Line 376: “… almost double the prevalence…”.

Is most of the double of each of the other two cities.

Lines 456-457: “…was relatively low, awareness of the benefits of opioid agonist therapy was low…”

This was not mentioned before in the article, can be specified where this come from?

6. PLOS authors have the option to publish the peer review history of their article (what does this mean? ). If published, this will include your full peer review and any attached files.

**Do you want your identity to be public for this peer review?** For information about this choice, including consent withdrawal, please see our Privacy Policy .

Reviewer #1: No

Reviewer #2: **Yes: ** Keith Sabin

---

## [Author Response · Author response to Decision Letter 1]

6 Mar 2025

We recommend accessing our attached document entitled "Response to Reviewers" where all of our responses are laid out neatly, but we pasted that document here as well.

"https://journals.plos.org/plosone/s/file?id=ba62/PLOSOne_formatting_sample_title_authors_affiliations.pdf"

Author Response: Thank you. We have gone through and checked, and updated the formatting of the author list to denote groupings of those who contributed equally.

Author Response: This file has been included with our resubmission. The file name is “Inclusivity-in-global-research-questionnaire”.

Author Response: The Award Number is: 1NU2GGH002056. We have ensured the reported numbers match in the financial disclosure statement and funding information section.

[This activity was supported by the US President’s Emergency Plan for AIDS Relief (PEPFAR) (https://www.state.gov/pepfar/) through the US Centers for Disease Control and Prevention (CDC) under the terms of cooperative agreement “Supporting sustainable surveillance systems among key populations and support the Government of Zambia to improve HIV-related services for KPs” (Prime Award No. 1NU2GGH002056). The findings and conclusions in this report are those of the author(s) and do not necessarily represent the official position of the funding agencies. ].

Author Response: Thank you. We have amended the above financial statement to also include the requested information, which has been included in its totality in the cover letter as requested. You can see the revised statement here as well:

“This activity was funded by the US President’s Emergency Plan for AIDS Relief (PEPFAR) (https://www.state.gov/pepfar/) through the US Centers for Disease Control and Prevention (CDC) under the terms of cooperative agreement “Supporting sustainable surveillance systems among key populations and support the Government of Zambia to improve HIV-related services for KPs” (Prime Award No. 1NU2GGH002056). The findings and conclusions in this report are those of the author(s) and do not necessarily represent the official position of the funding agencies. CDC employees, whose specific roles are outlined in the author contributions section, provided technical assistance and input into the study design, data collection, data analysis, decision to publish, and preparation of the manuscript.”

5. We note that you have indicated that there are restrictions to data sharing for this study. For studies involving human research participant data or other sensitive data, we encourage authors to share de-identified or anonymized data. However, when data cannot be publicly shared for ethical reasons, we allow authors to make their data sets available upon request. For information on unacceptable data access restrictions, please see http://journals.plos.org/plosone/s/data-availability#loc-unacceptable-data-access-restrictions. Before we proceed with your manuscript, please address the following prompts: a) If there are ethical or legal restrictions on sharing a de-identified data set, please explain them in detail (e.g., data contain potentially identifying or sensitive patient information, data are owned by a third-party organization, etc.) and who has imposed them (e.g., a Research Ethics Committee or Institutional Review Board, etc.). Please also provide contact information for a data access committee, ethics committee, or other institutional body to which data requests may be sent. b) If there are no restrictions, please upload the minimal anonymized data set necessary to replicate your study findings to a stable, public repository and provide us with the relevant URLs, DOIs, or accession numbers. Please see http://www.bmj.com/content/340/bmj.c181.long for guidelines on how to de-identify and prepare clinical data for publication. For a list of recommended repositories, please see https://journals.plos.org/plosone/s/recommended-repositories. You also have the option of uploading the data as Supporting Information files, but we would recommend depositing data directly to a data repository if possible. Please update your Data Availability statement in the submission form accordingly.

Author response: Thank you for this comment. The underlying data for this study are owned by the Zambian Ministry of Health and the National AIDS Council. Due to the sensitive nature of the data, which involve key populations that face legal and social risks, there are strict ethical and legal restrictions on data sharing. These restrictions are mandated by the institutional review board (IRB) requirements to protect participant privacy and confidentiality. For these reasons, the de-identified data can only be accessed upon request and after approval by the relevant Zambian health authorities. Researchers interested in accessing the data should contact:

Dr Kebby Chongwe Musokotwane

Director General

National HIV/AIDS/STI/TB Council

Plot 315, Independence Avenue

PO Box 38718

Lusaka, Zambia

Tel: +260-211-255044

KMusokotwane@nacsec.org.zm

Requests will be evaluated to ensure compliance with ethical standards and participant safety.

Author Response: Thank you. We have ensured our reference list is complete and correct.

Additional Editor Comments:

1. Introduction (lines 82 - 84). The objective does not mention assessment or presentation of results around syphilis, HBV or HCV (STI and BBV), so consider noting these too.

Author Response: Thank you for noticing this. We have added these components into the relevant lines of the Introduction Section of the revised manuscript.

2. Introduction. The provision and availability of needle and syringe programmes (NSP) and opioid agonist maintenance therapy (OAMT) and treatment of stimulant dependence in Zambia should be briefly described.

Author Response: Thank you. The following basic information has been added into the 3rd paragraph of the introduction section:

“In addition, there are no PEPFAR or government-sponsored Opioid Agonist Maintenance Therapy (OAMT) or Needle and Syringe Programs (NSPs) available for the treatment of stimulant dependence in Zambia.”

We have also added information to the discussion section about some work that is happening to create policy, guidance, and institutional space for the hopeful enabling of methadone treatment in Zambia. This addition to the discussion section has been explained in more detail in our response to the editor’s later comment about NSP and OAMT in #11 below.

3. Methods (lines 92). Insert a short description about each study site for readers who may not be familiar with Zambia.

Author Response: The following descriptions have been added to the methods section:

“Livingstone is in Southwest Zambia on the border with Zimbabwe and has a high influx of visitors for Victoria Falls, but nonetheless has the smallest population of the three survey sites. Lusaka is in Central Zambia, is the nation’s capital, and has the largest population of the three sites. Ndola lies roughly 300 km to the north of Lusaka near the border with the Democratic Republic of the Congo.”

4. Findings. The findings section (notably descriptions) is a bit long in parts. Aim to focus on key findings, and reduce some descriptions of socio demographic characteristics, and refer to the table.

Author Response: Thank you, we agree. The language throughout the whole ‘Results’ section has been streamlined and shortened while some of the repetitive/less important descriptions have been reduced or omitted.

5. Findings. Table 1. Insert the % for all rows, where relevant (currently only done for some).

Author Response: We agree this helps with the standardization of the table. This has been done.

6. Findings. Unless I missed it, I did not see the description of the type of drugs injected. This is critical to assess risk, prevention and interventions.

Author Response: The most common type of drugs injected across the three sites has been added to Table 3 and the corresponding paragraph in the findings. There will be another manuscript published from another partner research group that goes into more detail about injection drug practices, prevention, and interventions in this PWID population, which is why we originally didn’t include it here. However, we agree that this brief background will help contextualize the 95-95-95 progress we are reporting on. Thank you for this suggestion.

7. Table 3. Median duration of injecting. Check the median and p25 and p75 - I would expect the median to lie between these, but does not seem to be the case (e.g., Ndola median is 3.0 years, but IQR is 6 - 10 year).

Author Response: Thank you so much for noticing this, not sure how it slipped by after so many reviews on our end. These medians and their p25 and p75’s have been recalculated and have now been fixed in Table 3. These numbers have been corrected within paragraphs where median duration of injecting was reported or referenced as well.

8. Insert n for anti-HCV results and HCV PCR for sample in table and findings description.

Author Response: Thank you for your comment. With such small sample sizes, it was the team’s policy to put an asterisk in place of n’s that were 5 or less in order to protect this vulnerable population from being identified through data triangulation. In any tables in which an asterisk is shown, the asterisk is described as such in the table’s footnote. All of the local authors and advisors from the vulnerable groups themselves agreed with this decision. It was a deliberate choice so they could not be identified by an employer, family member, etc. since stigma and bias is so common with this group.

9. Findings around awareness of OAMT - this would only apply to people with opioid dependence, but it is not clear what drugs people injected. Also, Table 3 notes " aware of any drug treatment programs" - this is too broad to reflect insights into OAMT. So if questions on OAMT were asked, please include. If not, this could be noted in the limitations.

Author Response: Thank you, this is great insight. In response to this and your previous comment about not knowing which types of drugs people injected, we included in Table 3 the most common types of drugs injected across the three sites, so it is our hope this provides some helpful context. In addition, an asterisk is included in the ‘Awareness of any drug treatment programs’ variable of Table 3 corresponding to a footnote that provides further details by stating that the variable includes outpatient, inpatient, residential, detox, and methadone maintenance drug treatment programs. Unfortunately, the original survey question written years ago didn’t tease out awareness based on type of program into any more detail than this, so this has been noted in the Limitations section too as you suggested:

“Our variable on awareness of drug treatment programs (including outpatient, inpatient, residential, detox, and methadone maintenance therapy) is of limited value since OAMT and NSPs are currently not available in Zambia, however the low awareness observed indicates work must be done to increase awareness on the benefits of these programs if they are to be maximally successful if and when they are approved in Zambia.”

10. Findings. Why no findings of needle reuse/ sharing? This is the most important risk factor for PWID? (Unless I missed it). This is key to include, as it is a UNAIDS GAM indicator. Please include these variables.

Author Response: An indicator regarding frequency of needle reuse in the last 6 months has been included in Table 3 and the corresponding paragraph. We definitely agree with your statement but originally didn’t include this variable in the manuscript because survey designers wrote this question subjectively. The original survey question had response categories of ‘Never’, ‘Rarely’, ‘Half of the time’, ‘Most of the time’, ‘Always’, and ‘Don’t know/No answer’. For the purposes of the paper, these categories have been recoded into ‘Never’, ‘At least once’, and ‘No answer’ to make it as objective and accurate as possible given the constraints in how this question was originally constructed.

11. Discussion. Among PWID, the primary HIV prevention interventions are NSP and OAMT, these should be foregrounded. These interventions are more important for PWID than PrEP, yet there is much emphasis on PrEP. Similarly, the implications for policy and practice, NSP and OAMT need to be much higher priorities than PrEP, but the framing does not align with emphasis on the issues that are particular to PWID. There is mention of MAT clinics, but what about NSP? This is critical, as many African countries are resistant to NSP, but without NSP HIV prevention interventions for PWID will fail to be effective.

Author Response: We agree NSPs and OAMT are very important for PWID. Unfortunately, as discussed above in our response to your other comment regarding the introduction section, they are currently not available in Zambia, whereas PrEP is. However, there is work going on regarding getting OAMT approved so we have added some detail about this into our “Implications for Program and Policy” subsection of the Discussion. Please see the information that has been added to the discussion section below:

“…Preventive resources like PrEP can be efficiently offered in these facilities as well (59), along with OAMT (60)) and NSPs. Results from the 2021 BBS have already propelled the establishment of a PWID task force with representation from government and non-governmental agencies that is working to establish MAT clinics while the police and Home Affairs Ministry offer basic rehabilitati

---

## [Decision Letter · Decision Letter 1]

17 Apr 2025

Local data for local programming: Results from an HIV biobehavioral survey among people who inject drugs in Livingstone, Lusaka, and Ndola, Zambia, 2021

PONE-D-24-29546R1

Dear Dr. Woytowich,

We’re pleased to inform you that your manuscript has been judged scientifically suitable for publication and will be formally accepted for publication once it meets all outstanding technical requirements.

Kind regards,

Ibrahim Jahun, MD, MSC, PhD

Academic Editor

PLOS ONE

Additional Editor Comments (optional):

Reviewers' comments:

Reviewer's Responses to Questions

**Comments to the Author**

1. If the authors have adequately addressed your comments raised in a previous round of review and you feel that this manuscript is now acceptable for publication, you may indicate that here to bypass the “Comments to the Author” section, enter your conflict of interest statement in the “Confidential to Editor” section, and submit your "Accept" recommendation.

Reviewer #3: (No Response)

Reviewer #4: All comments have been addressed

2. Is the manuscript technically sound, and do the data support the conclusions?

Reviewer #3: Yes

Reviewer #4: Yes

3. Has the statistical analysis been performed appropriately and rigorously? 

Reviewer #3: Yes

Reviewer #4: Yes

4. Have the authors made all data underlying the findings in their manuscript fully available?

Reviewer #3: No

Reviewer #4: Yes

5. Is the manuscript presented in an intelligible fashion and written in standard English?

Reviewer #3: Yes

Reviewer #4: Yes

6. Review Comments to the Author

Reviewer #3: The manuscript presents a cross-sectional survey of people who inject drugs (PWIDs), aged over 16 years, on HIV indicators, syphilis, HBV and HCV and associated risk behaviours in three cities in Zambia: Livingstone, Lusaka and Ndola. The median age was 23 in Livingstone, 24 in Lusaka and 28 in Ndola. There was low female representation among PWIDs with Ndola having the highest at 30% and Lusaka lowest at 2%. The research shows significant differences in HIV prevalence of PWIDS across the three sites with the highest having a prevalence of 21.9% (Ndola) and the lowest 7.3% (Lusaka). Risk perception was low among all the three sites. Progress towards the UNAIDS 95/95/95 targets was poor among the three sites with Lusaka being the worst performing. Sypilis prevalence was highest in Ndola at 11.1% and lowest in Livingstone at 3%. Prevalence of both hepatitis B and C was low.

The authors have highlighted the study limitations that include social desirability and recall bias.

Overall, the manuscript is well written, with the various sections bringing out the relevant information. The presentation of the results by site/city has revealed important similarities and differences that could inform PWID program implementation.

The conclusions reflect the data that is presented.

The authors have not made all the data available due to ethical and legal reasons. They have described where the data can be accessed.

Reviewer #4: This recommendation I have made is based on authors response to the initial reviewers' comments. Having read the revised (clean) manuscript before reading the the reviewers comments, and authors response and tracked version, I was able to appreciate that the manuscript was clear and logical. Nothing stood out as a particular concern to me. Being from the country where this work was done, I am also able to relate with the context in which this survey was done and attest that this has been an under studied key population. Therefore, the data presented is highly relevant and highlights an important challenge for Zambia to achieving the 95-95-95 control of HIV by 2030. This challenge is certainly not unique to Zambia. Even though the manuscript highlights marked differences in the cities that were considered, the recommendations are widely applicable. The messages and recommendations contained in this manuscript may be applicable to other settings in the region and globally as it relates to PWID and HIV.

7. PLOS authors have the option to publish the peer review history of their article (what does this mean? ). If published, this will include your full peer review and any attached files.

**Do you want your identity to be public for this peer review?** For information about this choice, including consent withdrawal, please see our Privacy Policy .

Reviewer #3: No

Reviewer #4: No

---

## [Editor Report · Acceptance letter]

PONE-D-24-29546R1

PLOS ONE

Dear Dr. Woytowich,

I'm pleased to inform you that your manuscript has been deemed suitable for publication in PLOS ONE. Congratulations! Your manuscript is now being handed over to our production team.

Kind regards,

on behalf of

Dr. Ibrahim Jahun

Academic Editor

PLOS ONE